# Multilayered PVDF-HFP Porous Separator via Phase Separation and Selective Solvent Etching for High Voltage Lithium-Ion Batteries

**DOI:** 10.3390/membranes11010041

**Published:** 2021-01-07

**Authors:** Van-Tien Bui, Van-Toan Nguyen, Ngoc-Anh Nguyen, Reddicherla Umapathi, Liudmila L. Larina, Jong Heon Kim, Hyun-Suk Kim, Ho-Suk Choi

**Affiliations:** 1Department of Chemical Engineering and Applied Chemistry, Chungnam National University, Yuseong-Gu, Daejeon 34134, Korea; tienbv@cnu.ac.kr (V.-T.B.); toanvibm@gmail.com (V.-T.N.); Ngocanh888bk@gmail.com (N.-A.N.); umapathi4u@gmail.com (R.U.); llarina3333@gmail.com (L.L.L.); 2Department of Materials Science and Engineering, Chungnam National University, Yuseong-Gu, Daejeon 34134, Korea; mapig2@naver.com (J.H.K.); khs3297@cnu.ac.kr (H.-S.K.); 3Institute Faculty of Materials Technology, HoChiMinh University of Technology-VNU HCM, HoChiMinh City 10288, Vietnam; 4Department of Solar Photovoltaics, Institute of Biochemical Physics, Russian Academy of Sciences, Kosygin St. 4, 119334 Moscow, Russia

**Keywords:** multilayer membrane, microporous membrane, phase separation

## Abstract

The development of highly porous and thin separator is a great challenge for lithium-ion batteries (LIBs). However, the inevitable safety issues always caused by poor mechanical integrity and internal short circuits of the thin separator must be addressed before this type of separator can be applied to lithium-ion batteries. Here, we developed a novel multilayer poly(vinylidene fluoride-co-hexafluoropropylene) (PVDF-HFP) membrane with a highly porous and lamellar structure, through a combination of evaporation-induced phase separation and selective solvent etching methods. The developed membrane is capable of a greater amount of electrolyte uptake and excellent electrolyte retention resulting from its superior electrolyte wettability and highly porous structure, thereby offering better electrochemical performance compared to that of a commercial polyolefin separator (Celgard). Moreover, benefiting from the layered configuration, the tensile strength of the membrane can reach 13.5 MPa, which is close to the mechanical strength of the Celgard type along the transversal direction. The elaborate design of the multilayered structure allows the fabrication of a new class of thin separators with significant improvements in the mechanical and electrochemical performance. Given safer operation, the developed multilayer membrane may become a preferable separator required for high-power and high-energy storage devices.

## 1. Introduction

The development of porous polymer membranes has attracted remarkable attention in such fields as separation, energy storage, and crystallization [1,2,3]. Recently, much research endeavor has been dedicated to enhancing the thermal stability, wettability, and mechanical properties of membranes. Among polymers, polyvinylidene fluoride (PVDF) has been widely used in a large range of applications due to outstanding mechanical strength, excellent chemical resistance, good thermal stability, and high hydrophobicity [4,5]. However, the copolymer of PVDF and hexafluoropropylene (PVDF-HFP), possessed higher solubility, higher hydrophobicity, greater free volume, and better mechanical strength, has been prominently emerged as a fascinating membrane material because the incorporation of HFP groups enhances the fluorine content compared with original PVDF polymer [2]. Many attempts were made to enhance the mechanical properties of polymer separators by adding nanofillers, using cross-linkers, and employing nonwoven materials as mechanical supports [1,2].

Numerous studies have addressed to nanofillers used for enhancing the mechanical strength and increasing the ion conductivity of a composite membrane [6,7,8,9,10,11]. However, the uniform dispersion of nanomaterials into an organic-based polymer solution is difficult because of their high surface energy and large energy disparity compared to those of common organic solvents [6,9]. The aggregation of the nanomaterials neutralizes any nanoscale benefits [9]. Moreover, abundant functional polar groups appearing on nanomaterials may not be electrochemically stable against the strong oxidizing and reducing environments posed by the active electrodes under charge/discharge cycles, which can interfere with the functions of the battery.

An alternative approach offers a nonwoven mat employed as a thin support film [10]. Although this type of nonwoven mat can provide good mechanical strength and useful thermal shutdown properties, it causes an increment in the thickness and a reduction in the porosity of the composite separator [11,12]. These disadvantages limit the electrochemical performance and life cycle of high-power LIBs. Moreover, in some cases, prior to use nonwoven materials in producing a composite separator, additional prerequisite complex steps are required leading to an increase in production cost [11].

Cross-linking of the polymer matrix has also been suggested to improve the mechanical and thermal stability levels of polymer electrolytes. This concept can be an effective approach to produce highly flexible and expandable gel electrolytes [13,14,15,16]. However, the formation of a dense polymer phase in most cross-linked gel polymer electrolytes causes difficulty in filling with a liquid electrolyte. Moreover, it adversely affects the residual initiators on the battery compounds such as a liquid electrolyte, electrode materials, or even a polymer separator, leading to the degradation of the battery performance under cycling. To overcome the disadvantage mentioned above, we developed a unique layered configuration which offers a variety of potential benefits with which to address issues related to the mechanical integrity and electrochemical stability of gel polymer electrolytes made with a thin separator. First, such a configuration enhances the mechanical strength and flexibility of a polymer electrolyte because an interspace between the layers can alleviate large extrusions when applied with stress. Second, a stack of layers can suppress short circuits deriving from the low structural uniformity of each thin film. Third, the gap space can act as an electrolyte reservoir to facilitate high electrolyte adsorption, leading to improved battery performance.

To date, strict requirements for the efficient use of energy sources accelerate the investigation in the field of storing energy [17]. High voltage lithium-ion batteries (LIB) have attracted enormous amount of attention as one of the most promising energy storage system due to their high specific energy and long cycle lifetimes [18,19]. The studies have been focused on a major current issue—producing higher power/energy densities while maintaining the safe operation of LIBs [20,21]. The separator has intimate influences on battery operation, battery performance, and is finally the most prominent in the safety and dependability of LIBs [22,23,24]. The separator prevents physical contact between the positive and negative electrodes while serving as the electrolyte reservoir to facilitate ionic transport during the charging and discharging cycles of the battery. Therefore, the separator not only determines the safety but also the performance of LIBs influencing the cell kinetics [25,26,27,28,29]. With regard to the ion transporting capability, a separator should be highly porous to enable the adsorption of liquid electrolytes, thus supporting the high electrochemical performance of the battery. The porosity of separator is an important parameter due to its influence on the overall impedance and, in turn, the electrochemical performance of the battery cell [29]. A separator with higher porosity provides several potential advantages such as improvement of electrolyte wettability and thermal stability, higher energy density levels, faster ion transport, and longer cycling lifetimes [23,28,30,31]. However, separators with high porosity always have adverse effects including poor mechanical integrity and low structural uniformity while also introducing a high probability of puncture formation during the cell assembly. These issues lead to unreliable operation of LIBs. Therefore, the development of a new highly porous multilayer separator with superior mechanical and structural integrity is a major challenge. The PVDF-HFP membrane is one of the most 5 choices for LIB separator material because of its high dielectric constant, superior affinity to liquid electrolyte solutions, good electrochemical stability, and desirable adhesion with electrodes [25].

In this work, a novel method has been designed to prepare a highly porous PVDF-HFP membrane with a multilayered structure via the combination of evaporation-induced phase separation (EIPS) and selective solvent etching. Membranes with structures consisting of one, two, three, and four layers were synthesized. The multilayered membrane was controlled to be less than the thickness of the commercial Celgard separator. The characteristic properties of the membranes, such as the structural morphology, physical properties, and electrochemical performance capabilities, were characterized. The obtained parameters were compared to those of a reference separator, the commercial Celgard separator. The developed membrane demonstrated superior electrolyte wettability, high porosity, good mechanical properties, and thermal stability. The multilayered membrane yields excellent electrochemical performance due to its significant electrolyte uptake. The obtained results provide an efficient and reliable approach for the construction of high-power and high-energy storage devices.

## 2. Experimental Section

### 2.1. Materials and Chemicals

Poly(vinylidene fluoride-co-hexafluoropropylene) (PVDF-HFP, MW ~455,000, pellets), Poly(4-styrene sulfonic acid) (PSS, MW ~75,000, 18 wt% in H_2_O), Lithium hexafluorophosphate (LiPF6), ethylene carbonate (EC), and diethyl carbonate (DEC) were purchased from Sigma-Aldrich (Yongin-si, Korea). Polystyrene pellet (PS) (GPPS 15NFI) was purchased from LG Chemical Co. Ltd. (Daejon, Korea). The polymers were initially dried in a vacuum oven at 40 °C. The PSS was initially dried by a rotary evaporator and then re-dispersed in methanol to obtain a 1 wt.% solution. Acetone (99.9%), chloroform (anhydrous, 99.8%), and methanol (anhydrous; 99.8%) were purchased from Sigma-Aldrich (Yongin-si, Korea) and were used as received.

Microscope cover glasses (24 × 60 mm) were purchased from Sigma-Aldrich (Yongin-si, Korea). Copper substrates (0.5 mm thickness) were obtained from 4Science (Seoul, Korea). Prior to use, the substrates were washed successively with acetone, a diluted sulphuric acid solution, and distilled water followed by blow-drying under nitrogen gas. PET films with protective layers on both sides were kindly supplied by SKC Co. Ltd. of Seoul, Korea. The cover layers were removed prior to use. A lithium foil with a thickness of 0.2 mm and a diameter of 16 mm was purchased from Wellcos Corporation (Pyeongtaek-si, Korea). A commercial separator (PP, Celgard 2400, with a thickness of 27 µm) was purchased from Celgard Korea, Inc. (Cheonan-si, Korea) to serve as a control sample.

### 2.2. Fabrication Process of the Porous PVDF-HFP Film

PVDF-HFP (6 g) was dissolved in acetone (98 mL) at 40 °C to form a homogeneous solution. Subsequently, water (2 mL) was added dropwise into the solution under intensive agitation. The solution was then kept for 2 h under continuous stirring at 40 °C, followed by aging for 1 h before being used to prepare the porous film by a dip-coating process. Herein, we used a water-soluble PSS layer to facilitate the detachment of the thin film from the substrate. A thin PSS layer was formed by the bar coating of 1 wt% of the PSS solution (dissolved in methanol) onto the solid substrate. The porous polymer thin film was prepared by dip-coating the ternary solution onto the substrate using a dip-coater (E-flex, Seoul, Korea) (SI, Appendix A). The sample was then kept in the air to dry it. The porous polymer film was spontaneously formed after the complete evaporation of the solvents. The drying conditions including the relative humidity (RH) and temperature (T) was controlled using humidity and temperature controller (THTG-1000, JeioTech, Seoul, Korea) (SI, Appendix A). The optimization of the parameters for the dip-coating process results in 50 cm/min for dipping speed, 60 cm/min for withdrawal speed, and 10 s for dipping time [32,33]. After depositing the first layer of the porous PVDF-HFP, 5 wt.% of a polystyrene solution (dissolved in chloroform) was coated onto the as-prepared porous PVDF-HFP thin film by spin-coating at a rotational speed of 1000 rpm for 40 s. This sample is denoted as PS/PVDF-HFP-1. PS layer will act as a sacrificial layer. The second layer of porous PVDF-HFP was deposited onto PS/PVDF-HFP-1 sample using the same dip-coating procedure, and this sample denoted as PS/PVDF-HFP-2. Multilayered porous PVDF-HFP composite films were prepared by sequentially depositing porous PVDF-HFP onto the PS/PVDF-HFP layers.

Finally, the porous PVDF-HFP composite film was detached from the solid substrate by immersing it in water for several seconds. The PS sacrificial layer was removed from the composite film by a selective solvent etching method using chloroform. Because chloroform is a good solvent for PS but a poor solvent for PVDF-HFP, the PS could be completely removed by soaking the film in chloroform for 2 h while preserving the porous structure of the PVDF-HFP film.

### 2.3. Fabrication of Electrodes and Lithium Ion Battery Assembly

LiNi_0.5_Mn_1.5_O_4_ (LNMO) is chosen as a cathode material due to its high-voltage cathode, high energy density, and low environmental impact [34,35,36,37]. To prepare the LNMO cathode material, a lithium nickel manganese spinel LiNi_0.5_Mn_1.5_O_4_ thin film with a thickness of 400 nm was deposited onto a stainless steel (SS) substrate by Radio Frequency (RF) sputtering. The sputtering power was 1.75 W/cm^2^ and the flow rate of Ar gas was 60 sccm. After the coating process, the cathode was post-annealed at 600 °C for 2 h in a furnace at a heating rate of 10 °C/min. A 1 M of LiPF_6_ in a mixture of the EC and DEC with the volume ratio of 1/4 was used as an electrolyte.

Li/LNMO half cells (CR2032-type coin cells with a cathode diameter of 16 mm) were assembled by sandwiching a separator filled with liquid electrolyte between a lithium foil with a thickness of 0.2 mm as the counter electrode and active LNMO as the working cathode electrode. The coin cell assembly was carried out in an argon-filled glove box (KOREA KIYON, Seoul, Korea). After the assembly, the cells were kept in the box for 24 h for aging prior to testing.

### 2.4. Characterization and Measurements

The charge/discharge performance, C-rate discharge capability, and cycle life performance of the cells under study and the reference cell with Celgard separator were characterized. For this purpose, a battery testing system (WBCS 3000, WonATech, Seoul, Korea) was used. The charge/discharge test was conducted at cutoff voltages between 3.5 and 4.9 V, and the rate capability was assessed at rates which ranged from 0.1 to 2 C.

The morphology of the obtained porous film was examined using field-emission and high-resolution scanning electron microscopy (FESEM, HRSEM) (JEOL, JSM-7000F, Akishima, Tokyo, Japan). The surface features were quantitatively determined by measuring the SEM images using the Image J program. SEM images obtained under magnification of 3000× were selected because a sufficient number of pores for the characterization of the pore sizes and pore number densities are visible in the images. The wettability of the sample was characterized by measuring the static water contact angle using a drop shape analyzer (Krüss DSA 100, Hamburg, Germany). The crystallinity of the porous film was characterized by means of X-ray diffraction (XRD, Cu K- alpha, λ = 1.5406 Å; Bruker AXS, Germany). The chemical and molecular structures of the composite films were characterized using Fourier transform infrared spectroscopy (FTIR, Thermo Scientific, Waltham, MA, USA).

To measure the electrolyte uptake and electrolyte retention, a piece (2 × 2 cm^2^) of the dry membrane was soaked in a lithium-ion electrolyte for 2 h at room temperature. After soaking, the membrane was taken out of the liquid electrolyte and then was gently blotted with a paper tissue. The electrolyte uptake was calculated using the following equation:(1)$$\theta_{u}=\;\frac{W_{1}-\;W_{0}}{W_{0}}$$

The electrolyte retention was calculated as follows:(2)$$\theta_{r}=\;\frac{W_{1}-\;W_{0}}{W_{1}}$$
where *W*_0_ is the weight of the membrane before absorption of electrolyte and *W*_1_ is the weight of the membrane after absorption of electrolyte.

The mechanical properties of the membranes were characterized using a universal testing machine (Instron, Norwood, MA, USA). The test-samples were prepared with a width of 20 mm and a length of 40 mm. The tensile rate was found to be 5 mm min^−1^ and the gauge length was 20 mm. The tensile strength, *T*, was calculated as follows,
(3)$$T=\;\frac{P}{S}$$
where *P* is the tensile force applied to the sample at break and *S* is the cross-section area of the sample.

## 3. Results and Discussion

### 3.1. Phenomena and Formation Mechanism of the Multilayer Membrane

Scheme 1 illustrates the entire fabrication procedure used to create the multilayered porous PVDF-HFP film, which consists of the sequential coating of a PVDF-HFP layer and a PS layer onto the PSS-coated substrate.

In the initial step, the PSS solution was coated onto the substrate. Given that the detachment process is crucial to producing a free-standing film from a solid substrate, the PSS coating was used as a water-soluble sacrificial layer to facilitate the transfer process and to enable damage-free detachment. It was shown that polymer membranes with a through-pore structure can be formed on the hydrophilic surfaces, including water, ice, hydrophilic glass, and PSS coatings, whereas a closed one-end structure forms on a solid substrate [38,39,40,41]. Thereby, exploiting the PSS sacrificial layer not only facilitates detachment but also leads to the formation of numerous pores on the backside surface facing the substrate (SI, Appendix A). The higher pore density on the backside surface not only increases the porosity of the membrane but also minimizes the disparity of the porosity across the membrane, leading to a longer cycling lifetime and higher performance of the battery [29].

In the second step, the microporous thin PVDF-HFP film was deposited onto the surface of the PSS-coated substrate by the evaporation-induced phase separation (EIPS) using a simple dip-coating technique. EIPS is the simplest and the most effective method for making microporous separators [42]. Accordingly, a ternary solution containing PVDF-HFP, acetone, and water was dip-coated onto the substrate, followed by a normal air-drying step. Acetone and water were used correspondingly as a solvent and non-solvent pair. Note the water is an environmentally friendly and inexpensive pore inducer. The content of water relative to that of acetone can be as high as 5 vol.%. Because acetone has a higher evaporation rate than that of water, the water content in the solution gradually increases during the drying process. This causes phase separation, thereby giving rise to the porous structure after complete evaporation. Through this approach, the first layer of the PVDF-HFP porous film was generated.

Our previous experiences allow us to conclude that other coating methods are associated with several practical and technical problems during the preparation of porous polymer films. For instance, a porous thin film created by the spin-coating method can generate gradient pore sizes from the center to the boundary of the sample, and these films may not be suitable for mass production. With a bar coating, it can be quite difficult to create a thin film with high uniformity and reproducibility. Thus, here, we explored the advantages of the dip-coating method to generate a uniform thin film. Furthermore, this method allows controlling the thickness of the film by regulating the viscosity of the solution or the dipping conditions [32,33].

In the third step, the PS layer was deposited onto the first layer of the porous PVDF-HFP film by spin-coating a PS solution. In order to produce the next layer of the porous PVDF-HFP, two critical requirements must be fulfilled. First, the pre-prepared PVDF-HFP layer must be stable in the coating solution to become a support for the next PVDF-HFP layer. Second, the surface of the pre-prepared PVDF-HFP layer should be flat because it can affect the structural uniformity of the next PVDF-HFP layer. Therefore, a PS solution in chloroform was used in this work because PS neither dissolves in acetone nor in water but can be quickly dissolved in chloroform. In contrast, PVDF-HFP can be dissolved in acetone but cannot be dissolved in chloroform. Consequently, the porous PVDF-HFP film can be suitably preserved in the PS coating process. The PS coating plays key roles in this process, including preservation of the structure in the acetone solution and the creation of a flat surface for the coating of the next layer of the porous PVDF-HFP. The porosity and adhesion between two porous PVDF-HFP layers were determined by the internal space between the layers, which in turn was determined by the thickness of the PS coating. The PS coating thickness was controlled by regulating the viscosity of the PS solution and the rotational speed of the spin-coater (SI, Appendix A). It was found that the thickness of the PS coating should be suitable to provide a flat surface and high porosity after the extraction process. However, it should not be too large to ensure strong adhesion between the two layers.

In the fourth step, the second layer of the porous PVDF-HFP was deposited onto the surface of the PS layer using a dipping procedure identical to that used for the first PVDF-HFP layer. For the fabrication of membranes with a multilayer structure, subsequent coatings of the PS and PVDF-HFP solutions were done using the same procedure.

Finally, to detach the composite film from the substrate and to remove the intercalated PS layer selectively from the composite film, water and chloroform were used to dissolve the PSS and PS layers, respectively. Note that neither water nor chloroform can dissolve the PVDF-HFP polymer; thus, the porous structure of PVDF-HFP was well preserved after the detachment and extraction process.

### 3.2. Interfacial Properties of Porous PVDF-HFP Films 

Figure 1a shows an optical photograph of the PVDF-HFP membrane fabricated on a PSS/copper substrate using the EIPS method with a simple dip-coating technique. The membrane looks opaque because the light scatters off the microporous structure. The membrane shows a uniform structure over the coating area. 

XRD patterns of the smooth and porous PVDF-HFP films are given in Figure 1b. As indicated by the diffraction patterns of the smooth film, diffraction peaks observed at 2*θ* = 18.2, 20.9, and 24.3 correspond to the (100), (020) and (110) crystal planes of vinylidene fluoride (VDF) segments [6]. For the porous films, the relative intensity of the peaks is remarkably decreased compared to those of the smooth film. The decrease has appeared with an increase in the number of the PVDF-HFP layers. This result suggests that the formation of the porous structure induces structural reorganization of the PVDF-HFP polymer, leading to a significant decrease in its crystallinity and as consequence, an increase of the amorphous domain in the membranes. This characteristic feature facilitates higher liquid electrolyte absorption and swelling. These findings are supported by the electrolyte uptake capabilities.

Wettability is a critical property of the separator due to its influence on electrolyte retention and, thus on the performance and cycle lifetime outcomes of the LIBs. Figure 1c illustrates the optical images of the drop shapes of a liquid electrolyte (3 µL) on the commercial Celgard separator and the as-prepared porous PVDF-HFP film. As can be seen, the electrolyte drop remains upright on the Celgard sample even after 120 s, whereas it quickly penetrates within 5 s into the porous PVDF-HFP membrane (Video 1 in Supporting Information). The apparent contact angles of the Celgard sample with water and with a liquid electrolyte (EC/DEC = 1/4 *v*/*v* containing 1.0 M LiPF_6_) after 5 s are 106° and 68°, respectively, while the corresponding values for the porous PVDF-HFP are 135° and zero, respectively. In addition, the contact angle after 120 s was estimated to be 50°. These results indicate that the porous PVDF-HFP film has superior liquid electrolyte wettability while exhibiting high water-repellent capabilities. Thereby, the as-prepared porous PVDF-HFP membrane provides faster electrolyte absorption together with higher electrolyte retention but better moisture tolerance compared to the commercial Celgard sample, which is highly beneficial for developing advanced batteries. In addition, the fast absorption of the liquid electrolyte also facilitates the process of electrolyte filling, leading to easy assembly of the battery.

The SEM plain images of the surface of PVDF-HFP membrane facing air and of the bottom surface of PVDF-HFP film facing the PSS/copper substrate are given in Figure 1d,e. The surface morphology of the top and bottom sides of the PVDF-HFP membrane is very different. A uniform and highly porous honeycomb-like structure is observed for the top of the PVDF-HFP membrane. The pore diameter and the interval were found to be 4–6 µm and ~1.5 µm, respectively. In addition, many smaller pores are visible in subsurface range inside the micropores which were formed due to the interconnected structures of the multilayer pore array. Contrary, a shallow porous structure with a smaller pore size is observed on the bottom surface of the membrane (Figure 1e). As clearly seen, the distribution of the pores as well as the pore sizes are irregular. Witte et al. reported that the nonporous skin layers usually form on the surface of a membrane facing the substrate [42]. In order to suppress the adverse effects of the solid substrate on the formation of the porous structure, in our study, a water-soluble sacrificial PSS layer was bar-coated onto the substrate. We found that the PSS coating not only allows easy detachment of the thin polymer membrane from the substrate without any damage but also increases the pore density on the bottom surface (SI, Appendix A). The cross-sectional morphology shown in Figure 1f exhibits the multilayer interconnected symmetric porous structure of the pore array. The formation of a multilayer structure can be explained by the higher density of the non-solvent water compared to that of the acetone solution [39]. The thickness of the fabricated PVDF-HFP porous membrane was estimated to be 4.3 µm while the depth of the surface pore was ~2 µm (Figure 1f), which is very small when compared to those of commercial separators. Hence, its mechanical integrity is not sufficiently strong for its application as a separator in LIBs.

The PS/PVDF-HFP composite film was prepared by spin-coating a 5 wt.% PS solution in chloroform onto the porous PVDF-HFP film. The porous PVDF-HFP membrane was then recovered by selectively washing out the PS solution with chloroform. The effect of the washing process on morphology of the porous PVDF-HFP membrane is shown in Figure 2. It is seen that the porous PVDF-HFP membrane was covered with the PS coating and thereby the composite film demonstrated a relatively flat surface. However, the small holes can still be observed on the surface. Note that the existence of these holes is necessary to provide a strong connection between the two porous PVDF-HFP layers (SI, Appendix A). After immersing the composite film in chloroform for 30 min, the PS layer was removed from the PVDF-HFP surface while the porous structure of the PVDF-HFP layer was well preserved, as visible in Figure 2b.

Evidence of the complete wash-out of the PS from the composite film was confirmed by measuring the water contact angle (WCA) and an FTIR spectroscopy analysis of the composite film before and after the washing process. As can be seen in Figure 2c, the WCA of the composite film before removing the PS was 98°. However, the WCA reached 130° after removing the PS layer. This WCA value is very closed to that of the as-prepared membrane (129°). The lower WCA of the composite film before the PS removing can be attributed to the higher inherent hydrophilicity of PS and the lower surface roughness compared to those of the PVDF-HFP film. These results indicate that the PS was completely removed from the composite film and that the porous structure of the PVDF-HFP membrane was well preserved after the washing process. The FTIR spectra of three samples are shown in Figure 2d. The characteristic absorption peaks of PS including peaks at 1600, 1500, and 1400 cm^−1^, are recorded for the PS/PVDF-HFP composite film. These picks are attributed to the aromatic C-C bonds and the C-H bonding vibrations of the benzene ring. As can be seen in Figure 2d, the washing leads to the disappearance of PS characteristic peaks meaning that the PS layer was completely washed out from the composite film.

Thus, the multilayer composite film was fabricated through the in-series deposition of PVDF-HFP and PS layers. A multilayer porous PVDF-HFP membrane was obtained by the selective removal of the PS layers from the composite film. Figure 3 depicts the cross-sectional surface morphology of the composite films before and after the removal of the PS. As can be seen, the composite films before removal of the PS showed an upper porous PVDF-HFP layer and a densely structured PS/PVDF-HFP layer. The presence of a dense structure resulted from the impregnation of PS into the porous PVDF-HFP film. After removing the PS, the symmetric porous structure was noted throughout the film cross-section and a layered configuration was observed on the cross-section of the membranes. Interestingly, the thickness of the membrane after the washing process was found to be higher than that of the film before washing. We assume that the reorganization and the formation of the layer structure caused this phenomenon. As mentioned above, the layered structure of PVDF-HFP resulted from the selective etching of PS from the PS/PVDF-HFP. Thus, the spacing between the layers can be controlled by varying the PS coating thickness. Moreover, we found that the interface was clearly observable after the washing out of the thick PS coating (SI, Appendix A), suggesting the reducing of the risk of the short circuits, thus demonstrating an advance in the safety of LIBs.

### 3.3. Physical Properties of the Multilayered Porous PVDF-HFP Film 

The developed multilayered porous PVDF-HFP membranes were synthesized and characterized. The obtained results are listed in Table 1. The parameters of commercial Celgard Separator are also given for comparison. As can be seen, the thickness of the developed membranes was controlled to be less than 25 µm. The standard thickness of the commercial polyolefin separators such as the Celgard separator have a standard thickness estimated in the range of 20–30 µm. It is known that thick separators have high internal resistance, which decreases the cell performance [23,24,26]. Therefore, a minimal separator thickness is crucial for high-energy and high-power LIBs.

The high uptake amount of the liquid electrolyte plays a key role in improving the ionic conductivity and charge/discharge capability while high electrolyte retention is required for a long cycling lifetime of the LIB cell. Both the electrolyte uptake and the electrolyte retention amount of the layered membranes are much higher than those of the commercial Celgard separator (Table 1). The superior electrolyte uptake and electrolyte retention of the developed layered membrane are attributed to the combination of excellent electrolyte wettability, high porosity, and the unique layered structure of the membrane. Hence, by minimizing the thickness and improving the electrolyte absorption, the developed membrane can demonstrate rapid ionic transport, high ionic conductivity, and promising electrochemical performance when applied to LIBs.

High mechanical robustness of the separator is required when assembling the battery cell and to ensure safe battery operation. A tensile tester was used to characterize the mechanical properties of the developed membranes and the Celgard separator. The stress-strain outcomes are shown in Figure 4. Clearly, the tensile strength of the Celgard separator depends on the orientation, as it is produced via a uniaxial stretching process. The Celgard separator has a tensile strength value of 13.4 MPa along the transversal direction. The tensile strength of the layered membranes is close to that of the commercial separator along the transversal direction. It is important to note that high porosity can adversely affect the mechanical strength of these types of membranes; however, the membrane developed here with the layered structure not only has high porosity but also provides exceptional mechanical properties for normal lithium-ion battery applications. Importantly, the layer configuration is believed to enhance flexibility [24,26]. It should also be noted that the mechanical strength of the multilayer membrane is much higher than those of previously reported single-layer separators using the same PVDF-HFP material [43,44,45,46]. The inset in Figure 4 portrays images of the four-layer membrane and the commercial Celgard separator before and after a heat treatment at 150 °C for 30 min. As indicated, the Celgard separator did not withstand the elevated temperature well given that melted and formed a fiber-like structure. The high thermal shrinkage of the Celgard material at a high temperature causes internal short circuits between the electrodes. In contrast, the developed four-layer membrane demonstrated relatively good heat resistance with thermal shrinkage of less than 20% under identical conditions.

### 3.4. Cell Performance of the Fabricated PVDF-HFP Multilayered Porous Film

Figure 5a presents the comparable charge-discharge performance outcomes of the battery cells in the second cycle which was cycled in the cutoff voltage range of 3.5–4.9 V at a current density of 0.1 C. The discharge capacities of the battery cells assembled with the two-, three-, and four-layer membranes were 95.8, 101.5, and 89 mAh g^−1^, respectively, all of which were higher than discharge capacity of the commercial Celgard separator (87.1 mAh g^−1^). For a long-term stability test, cells composed of layered membranes and a commercial separator were tested with a C-rate of 1.0 C for 50 cycles, as shown in Figure 5b. As can be seen in Figure 5b, the initial discharge capacities of the cells assembled with the layered membranes were higher than that of the commercial Celgard separator. The capability retentions of the cells with the two-, three-, and four-layer membranes and the commercial Celgard separator were found to be 90, 95, 97, and 97%, respectively. Although cell with two-layer membrane demonstrated lower cycling stability, however, cells with the three-layer and four-layer membranes showed cycling stability comparable to a cell with a commercial separator. Figure 5c presents the discharge capability of cells assembled with the layered membranes and the commercial Celgard separator under various discharge conditions (0.1 C, 0.2 C, 0.5 C, 1 C, 2 C). As can be seen, all of the cells assembled with the layered membranes demonstrated higher discharge capacities than the capacity determined for Celgard separator at C-rates ranging from 0.1 to 2 C. The discharge capacity regularly dropped with an elevated discharge current density. However, the difference in the discharge capability between the layered membranes and the Celgard separator became more significant at higher discharge current densities. Among those samples, the cell with the three-layer membrane exhibited the best rate performance, corresponding to 89% of the capacity at the 0.1 C rate. The obtained results suggest that the three-layer membrane is a promising separator for high-power and rapidly charging LIBs with outstanding cycle lifetimes. The high capacities, as well as the excellent rate behaviors of the cells assembled with the layered PVDF-HFP membranes, are mainly due to the good interfacial compatibility with the electrodes, the highly microporous network, and the thinness of the cell. Such favorable parameters of developed membranes facilitate rapid transport of the ionic charge carriers for electrochemical reactions and provide the low internal resistance values at high current density levels.

## 4. Conclusions

Thin PVDF-HFP membranes with a highly porous and multilayered structure were designed and fabricated via a combination of evaporation-induced phase separation and selective solvent etching. The thicknesses of two-, three-, and four-layer PVDF-HFP membranes are approximately 13.5, 15.2 and 23.9 µm, respectively. The thicknesses of the membranes are considerably thinner than that of the commercial Celgard separator (27 µm) and the values reported in most works. The developed multilayered membrane demonstrated high porosity and superior electrolyte wettability, leading to a large electrolyte uptake of 228.5%. Moreover, due to the unique layered structure, the PVDF-HFP membranes possessed high mechanical strength along the transversal direction, which is close to that of a commercial Celgard separator. The thermal stability of the developed membranes was found to be better as compared to the Celgard separator. Considering the electrochemical properties, the LIB cell adopting the three-layer membrane exhibited the highest capacity, the best rate capability, and compatible cycling stability, making it a preferable thin separator for high-power (fast charging) LIBs with superior cycle performance and high safety.

In future work, enhanced performance and improved safety of layered PVDF-HFP separators can be acquired through precise control of the thickness of each PVDF-HFP and PS layer. The obtained results can be used in the strategy in the engineering of separators for application in high-power LIBs. The feasibility of these materials to other potential applications such as actuators and supercapacitors can be considered.

## Data Availability

Not applicable.

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
