# Peer review of "Multilayered PVDF-HFP Porous Separator via Phase Separation and Selective Solvent Etching for High Voltage Lithium-Ion Batteries"

_membranes, 2021, doi:10.3390/membranes11010041_

Round 1

Reviewer 1 Report

The manuscrept deals with the preparation and characterizzation of multilayered PVDF-HFP as polymer electrolyte for Lithium-ion batteries. The research is complete, weel designed and the conclusions are effectively supported by the experimental results.

I only have some minor concerns/curiosity:

1) how do the authors investigated the electrolyte retention capacity? I mean, did they left the electrolyte to equilibrate at a specific temperature and evaluated the mass variation before and after the treatment? Or they just calculated it right after the swelling in the liquid electrolyte?

2) It is quite interesting the thikness variation for each additional layer in the PVDF-HFP membrane. Indeed, in three-layer films the increase in the thikness was only 2.1 um, wherease was 8.4 in the four-layer film. How do the authors can explain the outcome?

Author Response

Reviewer1:

General Response: We would like to thank for your efforts to review our manuscript. We were greatly encouraged by your insightful and detailed comments to improve our manuscript. Our answers and comments are inserted right after your comments. The correspondingly added or modified texts in the revised manuscript are marked with yellow color.

1) how do the authors investigated the electrolyte retention capacity? I mean, did they left the electrolyte to equilibrate at a specific temperature and evaluated the mass variation before and after the treatment? Or they just calculated it right after the swelling in the liquid electrolyte?

Response

We investigated the electrolyte retention capacity follows the process:

To measure electrolyte retention, a piece (2×2 cm2) of the dry membrane was soaked in a lithium ion electrolyte for 2 h at room temperature. After soaking, the membrane was taken out of the liquid electrolyte and then was gently blotted with a paper tissue.

The electrolyte retention was calculated as follows:

     where W0 is the weight of the membrane before absorption of electrolyte and W1 is the weight of the membrane after absorption of electrolyte.

→ They just calculated it right after the swelling in the liquid electrolyte

2) It is quite interesting the thikness variation for each additional layer in the PVDF-HFP membrane. Indeed, in three-layer films the increase in the thikness was only 2.1 um, whereas was 8.4 in the four-layer film. How do the authors can explain the outcome?

Response

That's a good question. As mentioned below in the description of Fig. 3, it is thought that there is a tendency to thicken the layer structure by swelling due to the action of the lipophilic solvent used in the process of removing the PS sacrificial layer. In addition to this effect, it is a very difficult technology to form a multi-layered structure through the process of coating the PS sacrificial layer several times and removing it at once. Therefore, it seems that the difference in experimental reproducibility that may occur in this process cannot be ignored.

“The presence of a dense structure resulted from the impregnation of PS into the porous PVDF-HFP film. After removing the PS, the symmetric porous structure was noted throughout the film cross-section and a layered configuration was observed on the cross-section of the membranes. Interestingly, the thickness of the membrane after the washing process was found to be higher than that of the film before washing. We assume that the reorganization and the formation of the layer structure caused this phenomenon.”

Reviewer 2 Report

The paper of Bui et al. entitled “Multilayered PVDF-HFP Porous Separator via Phase 2 Separation and Selective Solvent Etching for High 3 Voltage Lithium-Ion Batteries” describes the preparation and the study of a thin multilayered separator to be potentially used in high-energy and high-power LIBs.

The work is well written, the methodology is thorough, the results well-presented and discussed, and the conclusions are well supported by the experimental findings. I find the work overall interesting and matching the Journal’s profile.

I found a few typos listed below. Please go through the manuscript once again.

A few comments are as follows:

Line 78. “amounts” à “amount"

Line 96. “most [..]” An adjective is missing.

Line 97. Erase “a”

Line 113. “pellet form”.

Lines 113, 114. “Mw” should be “MW”

Line 158. Stainless steel should be SS. Please define RF.

Lines 175-176. You started a new paragraph by mistake.

Line 187 and below. Please enumerate the equations sequentially.

Lines 271-272. Please check the sentence again.

Figure 1c. Which commercial liquid electrolyte did you use?

Line 307. There’s a full stop instead of a comma.

Table 1. I wouldn’t personally write the electrolyte uptake in %, as it is generally above 100%. Instead of writing, for instance, 196%, you could just write 1.96. It would be already understandable from equation 1 that the weight has almost tripled.

Figure 5. Were the tests repeated to assure reproducibility?

Discussion on figure 5 and conclusions. Do you have any idea why the 3-layered separator works the best? You could address this, even speculatively, proposing an idea.

Author Response

General Response: We would like to thank for your efforts to review our manuscript. We were greatly encouraged by your helpful comments to improve our manuscript. Our answers are inserted right after your comments. The correspondingly added or modified texts are marked with yellow color in the revised manuscript.

Line 78. “amounts” à “amount"

Response

We already corrected erased “amounts” to “amount"

Line 96. “most [..]” An adjective is missing.

Response

We think that this sentence “The PVDF-HFP membrane is one of the most 5 choices for LIB separator material because of its high dielectric constant, ….” is not wrong.

Line 97. Erase “a”

Response

We already erased “a”

Line 113. “pellet form”.

 Response

We corrected to polystyrene pellet

Lines 113, 114. “Mw” should be “MW”

Response

We already changed “Mw” to “MW”

Line 158. Stainless steel should be SS. Please define RF.

Response

 We corrected Stainless steel into SS and define RF as Radio Frequency

Lines 175-176. You started a new paragraph by mistake.

Response

 We already corrected the mistake

Line 187 and below. Please enumerate the equations sequentially.

Response

We already enumerate the equation as reviewer’s suggestion.

Lines 271-272. Please check the sentence again.

Response

We corrected the sentence “The membrane looks opaque because of the light scatters off the microporous structure. The membrane showed a uniform structure over the coating area.” into “The membrane looks opaque because the light scatters off the microporous structure. The membrane shows a uniform structure over the coating area.”

Figure 1c. Which commercial liquid electrolyte did you use?

Response

 We Lithium hexafluorophosphate (LiPF6), ethylene carbonate (EC), and diethyl carbonate (DEC). the commercial electrolyte was prepared as following a 1 M of LiPF6 in a mixture of the EC and DEC with the volume ratio of 1/4 was used as a liquid electrolyte.

Line 307. There’s a full stop instead of a comma.

Response

We corrected this mistake

Table 1. I wouldn’t personally write the electrolyte uptake in %, as it is generally above 100%. Instead of writing, for instance, 196%, you could just write 1.96. It would be already understandable from equation 1 that the weight has almost tripled.

Response

We corrected Table 1 as follows:

Table 1. Physical properties of the layered porous PVDF-HFP membranes and commercial Celgard separator.

Sample

Thickness

(µm)

Electrolyte uptake

Electrolyte retention

One layer

4.3

1.96

0.66

Two layers

13.5

2.28

0.70

Three layers

15.2

1.57

0.61

Four layers

23.9

1.48

0.60

Celgard

27.0

0.81

0.45

Figure 5. Were the tests repeated to assure reproducibility?

Discussion on figure 5 and conclusions. Do you have any idea why the 3-layered separator works the best? You could address this, even speculatively, proposing an idea.

Response

It is a very difficult technology to form a multi-layered structure through the process of coating various multilayers together with PS sacrificial layers several times and removing the PS layers at once. Therefore, it seems that the difference in experimental reproducibility that may occur in this process cannot be ignored. About why the 3-layered separator works the best has been already explained as follows; “The obtained results suggest that the three-layer membrane is a promising separator for high-power and rapidly charging LIBs with outstanding cycle lifetimes. The high capacities, as well as the excellent rate behaviors of the cells assembled with the layered PVDF-HFP membranes, are mainly due to the good interfacial compatibility with the electrodes, the highly microporous network, and the thinness of the cell.”

For the further specific answer to the reviewer's question, as we mentioned in the conclusion, we are currently studying how enhanced performance and improved safety of layered PVDF-HFP separators can be acquired through precise control of the thickness of each PVDF-HFP and PS layer.
